Translation, adaptation, validation and performance of the American Weight Efficacy Lifestyle Questionnaire Short Form (WEL-SF) to a Norwegian version: a cross-sectional study

Flølo Tone N. 1 2 tone.flolo@helse-bergen.no
Andersen John R. 3 4
Nielsen Hans J. 1
Natvig Gerd K. 2
1 Voss Hospital, Haukeland University Hospital, The Western Norway Region Health Authority , Voss, Bergen , Norway
2 Department of Global Public Health and Primary Care, University of Bergen , Bergen , Norway
3 Faculty of Health Studies, Sogn og Fjordane University College , Førde , Norway
4 Department of Surgery, Førde Central Hospital , Førde , Norway
Niaura Raymond
Electronic publication date: 2014 Sep 16
Publication date: 2014
Volume: 2
Electronic Location ID: e565
Received 2014 Jun 26; Accepted 2014 Aug 18
Copyright: © 2014 Flølo et al.
Copyright year: 2014
Copyright holder: Flølo et al.
License: This is an open access article distributed under the terms of the Creative Commons Attribution License, which permits unrestricted use, distribution, reproduction and adaptation in any medium and for any purpose provided that it is properly attributed. For attribution, the original author(s), title, publication source (PeerJ) and either DOI or URL of the article must be cited.
License URL: https://creativecommons.org/licenses/by/4.0/

Keywords: Eating self-efficacy, Morbid obesity, Bariatric surgery, Weight Efficacy Lifestyle Questionnaire, Self-efficacy

Funding: The authors declare there was no funding for this work.

==============================
Background. Researchers have emphasized a need to identify predictors that can explain the variability in weight management after bariatric surgery. Eating self-efficacy has demonstrated predictive impact on patients’ adherence to recommended eating habits following multidisciplinary treatment programs, but has to a limited extent been subject for research after bariatric surgery. Recently an American short form version (WEL-SF) of the commonly used Weight Efficacy Lifestyle Questionnaire (WEL) was available for research and clinical purposes.

Objectives. We intended to translate and culturally adapt the WEL-SF to Norwegian conditions, and to evaluate the new versions’ psychometrical properties in a Norwegian population of morbidly obese patients eligible for bariatric surgery.

Design. Cross-sectional

Methods. A total of 225 outpatients selected for Laparoscopic sleeve gastrectomy (LSG) were recruited; 114 non-operated and 111 operated patients, respectively. The questionnaire was translated through forward and backward procedures. Structural properties were assessed performing principal component analysis (PCA), correlation and regression analysis were conducted to evaluate convergent validity and sensitivity, respectively. Data was assessed by mean, median, item response, missing values, floor- and ceiling effect, Cronbach’s alpha and alpha if item deleted.

Results. The PCA resulted in one factor with eigenvalue > 1, explaining 63.0% of the variability. The WEL-SF sum scores were positively correlated with the Self-efficacy and quality of life instruments (p < 0.001). The WEL-SF was associated with body mass index (BMI) (p < 0.001) and changes in BMI (p = 0.026). A very high item response was obtained with only one missing value (0.4%). The ceiling effect was in average 0.9 and 17.1% in the non-operated and operated sample, respectively. Strong internal consistency (r = 0.92) was obtained, and Cronbach’s alpha remained high (0.86–0.92) if single items were deleted.

Conclusion. The Norwegian version of WEL-SF appears to be a valid questionnaire on eating self-efficacy, with acceptable psychometrical properties in a population of morbidly obese patients.

Introduction

Bariatric surgery is a well-established and approved treatment for patients suffering from morbid obesity (Colquitt et al., 2009). Increasing request for surgical treatment entails the epidemic dimension of morbid obesity as a worldwide public health threat (WHO, 2013). The magnitude of obesity is also present in a Norwegian context (Midthjell et al., 2013).

Bariatric procedures show excellent short term results (Karlsen et al., 2013; Andersen, 2011), and acceptable long term results (Sjostrom, 2013) with weight loss, remission of comorbidities and quality of life as the outcome measures. Nevertheless, between 30 and 40% of morbidly obese patients undergoing bariatric surgery seem to experience insufficient weight loss or regain weight (Biron et al., 2004; Magro et al., 2008; Livhits et al., 2012). Present researchers emphasize the need to identify predictors of sustained weight loss after bariatric surgery (Colquitt et al., 2009; Livhits et al., 2012). Changing old eating habits is, for some of the post-operative patients, reported to be a persisting challenge (Kafri et al., 2011). In order to offer suitable behavioral treatment for potential psychosocial obstacles in bariatric patients, it seems crucial to survey the impact of specific self-management skills.

Self-efficacy is a key concept in social cognitive learning theory (Conn et al., 2001; Bandura, 1997), and has by large demonstrated a predictive impact on individuals’ motivation and capability toward sustained behavioral change (Batsis et al., 2009; Bock et al., 1997; Condiotte & Lichtenstein, 1981). The concept refers to a person’s confidence in his or her ability to perform specific behavior in the face of perceived obstacles or challenging situations (Bandura, 1977). Even though self-efficacy has demonstrated significant prediction related to change in addictive behaviors, such as tobacco- and alcohol dependence (Condiotte & Lichtenstein, 1981; Trucco et al., 2007), and in exercise (Sallis et al., 1988; Sullum, Clark & King, 2000), the term was only recently applied to bariatric patients in their attempts on weight loss maintenance (Batsis et al., 2009).

According to social cognitive learning theory, individuals with low eating self-efficacy will have difficulties in resisting temptation to overeat in many situations (Cargill et al., 1999). Several studies have demonstrated that eating efficacy changes over time and improvements are associated with greater weight loss after multidisciplinary treatment programs (Bas & Donmez, 2009; Martin, Dutton & Brantley, 2004; Clark et al., 1996). Specific interventions performed to increase eating self-efficacy also obtained superior results in terms of weight management (Schulz & Mcdonald, 2011; Warziski et al., 2008). Furthermore, Batsis et al. (2009) demonstrated that profound weight loss after bariatric surgery was associated with increased long-term eating self-efficacy in post-bariatric surgery patients when compared with obese non-surgery patients. With regard to maintaining adherence to a recommended eating plan, self-efficacy therefore appears to be an important predictor (Batsis et al., 2009; Linde et al., 2006).

Research on eating self-efficacy is primarily based on global self-reported questionnaires such as the Eating Self-Efficacy Scale (ESES) (Burmeister et al., 2013; Pinto et al., 2008; Glynn & Ruderman, 1986). This instrument demonstrated acceptable psychometrical properties and produced preliminary support for self-efficacy theory in obesity treatment (Glynn & Ruderman, 1986). According to the authors, the predictive validity of ESES in a clinical setting would require further research (Glynn & Ruderman, 1986). Later researchers suggested that findings based on the ESES were limited due to the use of small, non-clinical samples in addition to incomplete psychometric methodology (Clark et al., 1991). By developing the Weight Efficacy Lifestyle Questionnaire (WEL), the authors extended previous studies on eating self-efficacy using a large sample (total N = 382) of obese persons examining treatment-produced change in two separate samples to explore the best fitting theoretical model of self-efficacy (Clark et al., 1991).

Patients selected for bariatric surgery are exposed to lengthy clinical assessments, and inclusion of further extensive measurements may be a burden for these patients. To address this challenge, Ames et al. (2012) developed a brief version of the WEL, labeled WEL-SF. A cross sectional validation study indicated that the short version captured 94% of the variability in the original WEL (Ames et al., 2012). Several studies indicate, accordingly, that well designed brief measures can be as valid as extensive ones (Marcus et al., 1992; Kolotkin et al., 2001; Clark et al., 2007).

The aim of this study was (1) to translate and adapt the WEL-SF to Norwegian conditions and (2) to test the new version’s psychometric properties in a Norwegian population of morbidly obese bariatric patients. A fourfold research question guided the study performance: (a) Is the WEL-SF a reliable questionnaire for eating self-efficacy? (b) Is the WEL-SF positively correlated with the General self-efficacy scale, the Self-efficacy for physical activity questionnaire, the SF-36 and the Impact of Weight on Quality of Life – Lite Questionnaire? (c) Does the WEL-SF hold an adequate structural robustness? (d) Does the WEL-SF perceive the different eating patterns between non-operated and operated patients?

Methods

Design, respondents and setting

The present study was conducted with a cross-sectional design including 225 morbidly obese patients accepted for bariatric surgery with laparoscopic sleeve gastrectomy (LSG) in a Western Norwegian hospital. We included two subsamples in the study; 114 consecutive non-operated patients from pre-operative outpatient consultations, and 111 consecutive operated patients from outpatient consultations one year after surgery, all within the period from October 2012 to May 2013.

The outpatient consultations started with a multidisciplinary informative plenary meeting, wherein the patients were shortly introduced to the present study. Voluntary participation was emphasized. Written information about the study was distributed with the questionnaires. Informed consent was obtained, and the questionnaires were collected at the end of the day before the respondents left the hospital.

The inclusion criteria were morbidly obese patients eligible for LSG (BMI ≥ 40, or ≥ 35 with comorbidity) and age between 18 and 60 years. Patients were excluded if they were physically or mentally disabled and could not fill in the forms.

Translation and adaptation (aim 1, research question a)

According to the recommendations in the guidelines by the World Health Organization (WHO), we performed a five step, systematic approach to translation and adaptation of the questionnaire (WHO, 2007). Initially, two registered dietitians who are native speakers of Norweigian and professionally familiar with the concepts about morbidly obese patients did an independent forward translation of the WEL-SF from American-English to Norwegian. Next, a consensus panel of four people comprised of the research group compared the original version with the two translated versions. The group reconciled the forward translations into one common version by identifying inadequate concepts or expressions. Third, two blinded backward translations into English were done by a surgeon and health educator, both of whom were native speakers of Norweigian. Furthermore, the consensus panel compared the original version and the translated version with respect to conceptual- and cultural equivalence and agreed on a Norwegian version for pretesting. Finally, two nurses, a registered dietitian and a bariatric surgeon, were asked to assess the feasibility of the items in the Norwegian version for the bariatric patients. They found the questionnaire to be of clinical relevance for the population.

The Weight-Efficacy Lifestyle Questionnaire Short Form (WEL-SF)

In 2012 a short version of the original WEL (Clark et al., 1991) was developed — from 20 questions and 5 situational components to 8 questions and 1 situational component representing “confidence in ability to resist eating” (Ames et al., 2012). Three of the questions are related to emotional eating situations, two to availability, one to social pressure, one to positive activities and one to physical discomfort. The WEL-SF correlated highly significantly with the WEL, accounting for 94% of the variability in the original questionnaire, and was found to be a psychometric valid measure of eating self-efficacy (Ames et al., 2012). The instrument range scores on a Likert-scale from 0 (not at all confident) to 10 (very confident), with sum scores between 0 and 80. High scores are associated with high eating self-efficacy.

Validating instruments

The instrument selection was based on a theoretical expected association with eating self-efficacy (Fayers & Machin, 2007). Weight loss maintenance after bariatric surgery requires a balance between energy intake and energy expenditure. It has been stated that this demands self-management skills in both eating behavior and physical activity (Sallis et al., 1988; Morin, Turcotte & Perreault, 2013; Wing et al., 2001). Due to this, we obtained the Self-efficacy for Physical Activity Questionnaire (SEPA) (Fuchs & Schwarzer, 1994) as one of the validating instruments. Based on social cognitive learning theory, we also assumed that individuals with high efficacy levels toward challenging life obstacles in general would be more likely to report high confidence in adequate manners of eating (Bandura, 1977; Sherer & Maddux, 1982). Thus, the General Self-efficacy Scale (GSE) (Luszczynska, Scholz & Schwarzer, 2005) served as a second validating measure. Furthermore, as the outcome expectations and measures of success in bariatric surgery is sustained weight loss and health related quality of life, we wanted to calculate the association between eating self-efficacy (WEL-SF) and health related quality of life, both in general and weight specifically. For this purpose the Short Form 36 (SF-36) (Ware, 2000) and the Impact of Weight on Quality of Life-Lite (IWQOL-Lite) (Kolotkin et al., 2001) were chosen as the third and fourth validating instruments.

Self-efficacy for physical activity (SEPA)

Self-efficacy for physical activity refers to the belief of being capable to stick to an exercise program even under unfavorable circumstances. The questionnaire was first developed in German by Fuchs & Schwarzer (1994) and assesses self-efficacy for physical activity using a 12-item measure on a Likert-scale ranging from 1 (very confident) to 7 (not confident at all). The sum score ranges from 12 to 84. High scores indicate high levels of perceived physical self-efficacy. The instrument was positively correlated with general self-efficacy and with specific self-efficacy expectations toward cancer screening and healthful eating behavior (Fuchs & Schwarzer, 1994), and has been translated and adapted to Norwegian conditions (Jenum et al., 2003).

General self-efficacy scale (GSE)

The General self-efficacy scale (GSE) contains general questions measuring an individual’s confidence in his or her personal competence to fulfill difficult tasks (Luszczynska, Scholz & Schwarzer, 2005). The instrument measures a person’s ability to cope with a broad range of demanding unspecific situations in life, and thereby assess his or her optimistic self-belief toward difficulties in general. The questionnaire has been translated, psychometrical tested and adopted for studies worldwide (Schwarzer et al., 1997; Scholz et al., 2002; Røysamb, Schwarzer & Jerusalem, 1998). The GSE contains 10 items on a Likert-scale ranging from 1 (completely wrong) to 5 (completely right). The sum scores ranges from 10 to 40. High scores indicate high levels of general self-efficacy.

Short form 36 (SF-36)

SF-36 is the most widely used generic self-report health questionnaire, which is based on a multidimensional model of health (Ware, 2000). The scale assesses health related quality of life outcomes, known to be most directly affected by unspecific disease and treatment and was first translated and adapted to Norwegian in 1998 (Loge et al., 1998). The 36 items are measuring 8 different aspects (subscales) of health related quality of life. The 8 subscale scores can be summed into two domains: physical component sum score (PCS) and mental component sumscore (MCS). The sub scores are transformed into a scale where high scores indicate high health-related quality of life. A score = 50 represents the average PCS and MCS scores in the US population. The psychometric properties of the SF-36 are well documented (Ware, 2000) and are validated for use in a Norwegian morbidly obese population (Karlsen et al., 2011).

Impact of weight on quality life-lite (IWQOL-lite)

Impact of Weight on Quality of life-lite is a validated, 31-item self-report measure of obesity-specific quality of life (Kolotkin et al., 2001). The questionnaire consists of a total score and scores on each of five scales; physical function, self-esteem, sexual life, public distress and work—exhibiting strong psychometric properties (Kolotkin et al., 2001). The subscores are transformed into a scale from 0–100 where high scores indicate high obesity specific quality of life. The version in use is linguistically-, but not yet psychometrically, validated in a Norwegian morbidly obese population.

Socio-demographic and clinical data

Socio-demographic variables of age, gender, marital status, level of education and work participation were recorded. The clinical variables include initial weight, weight loss, BMI, changes in BMI, height, diabetes, hypertension, psychiatric disorder, muscular- and skeletal pain and weather the respondents had undergone surgery or not. Changes in BMI were collected retrospectively. Data were coded and registered as categorical or continuous variables.

Statistical analysis

Data are presented as mean and standard deviation (SD) or number (%) unless otherwise stated. Between-group comparisons at baseline were analyzed using independent samples t-test for continuous variables and Pearson’s chi-square test for categorical variables. We employed two-tailed tests and considered P values <.05 statistically significant. The statistical analysis was conducted using SPSS version 21.0.

Internal validity (aim 2, research question a and c)

Data quality was examined comparing mean values for each item with standard deviation, median, percentage of missing values and extent of ceiling and floor effects. Optimal floor- and ceiling effects were defined to stay between 1% and 15% (Mchorney & Tarlov, 1995). Internal consistency was assessed by calculating Cronbach’s alpha coefficients. According to Clark & Watson (1995) the alpha coefficient should be benchmarked at .80 to raise reliability to an acceptable level. To eliminate the risk of a potentially false high reliability coefficient, we also calculated alpha if single items were deleted (Polit & Beck, 2008). Further, we measured the internal item convergence in terms of each items’ correlation with the rest of the scales’ total score. A minimum item-total correlation was benchmarked at the level of .3 (Fayers & Machin, 2007). In order to examine face validity, nine bariatric patients, included from the outpatient consultations one year after surgery, evaluated the questionnaire. The scales feasibility was assessed by four professional health workers.

Construct validity and factor analysis (aim 2, research question c)

To examine the structural validity of the WEL-SF we applied principal component analysis (PCA) with a varimax rotation (Tabachnick & Fidell, 2006). The Kaiser–Meyer–Olkin measure and Bartlett’s test of spherity were computed to determine whether the data in this sample were suitable for PCA. Following Kaiser’s criterion, eigenvalues of 1.0 were chosen to ensure that the extracted components accounted for a reasonably large proportion of the total variance (Tabachnick & Fidell, 2006). The PCA was first applied on the total sample (n = 225) and then on each subsample to compare the component structure between samples.

Convergent validity (aim 2, research question b)

Convergent validity was tested by comparing Pearson correlation coefficients between the WEL-SF and SF-36, the Impact of Weight on Quality of Life (IWQoL-Lite) questionnaire, the Self-Efficacy for Physical Activity Questionnaire (SEPA) and the General Self-Efficacy Scale (GSE).

Sensitivity (aim 2, research question a and d)

Multiple linear regression analyses were conducted to evaluate whether the WEL-SF discriminated between non-operated and operated patients, adjusted for age, gender, work participation, marital status and education.

Ethical approval

The study was approved by the Regional Committee for Medical Research Ethics in Western Norway and performed in accordance with the Helsinki Declaration (Saksnr 2012/1481).

Results

Characteristics of the participants

We included 225 Caucasian morbidly obese patients (69.3% women) accepted for bariatric surgery; 114 patients prior to surgery and 111 patients one year post-surgery. All patients that were asked to take part in the study agreed to participate, giving a response rate of 100%. A very high item response was obtained with missing values of only 0.4%. The missing items were not substituted. The distribution of answers was right-skewed with no floor effect and a ceiling effect of 8.9% for the entire sample; respectively 0.9% and 17.1% for the non-operated and operated subsample. A further characteristic of the respondents and description of data is shown in Tables 1–2 and Fig. 1.

Figure 1 Histogram.

Illustration of reported eating self-efficacy in the subsamples.

Table 1 Characteristics of the respondents.

Morbidly obese patients (N = 225).

	All patients	Non operated patients	Operated patients	
	(N = 225)	(N = 114)	(N = 111)	p a	
Age	42.5 (11.0)	41.9 (11.4)	42.9 (10.5)	0.47	
Female	156 (69.3%)	76 (66.7%)	80 (72.1%)	0.37	
Marital status				0.52	
Single	75 (37.8%)	43 (37.7%)	42 (37.8%)		
Partners	140 (62.2%)	71 (62.3%)	69 (62.4%)		
Education				0.61	
Primary/High	173 (76.9%)	88 (77.2%)	65 (76.6%)		
Bachelor/Master	52 (23.1%)	26 (22.8%)	26 (23.4%)		
Non employed	66 (29.3%)	39 (34.2%)	27 (24.3%)		
Initial BMI	43.2 (4.9)	42.7 (4.6)	43.8 (5.1)		
Comorbidities and self-reported measures					
Diabetes	30 (13.3%)	25 (21.9%)	5 (4.5%)	<0.001	
Hypertension	57 (25.3%)	37 (32.5%)	20 (18.0%)	0.01	
Psychiatric disorder	44 (19.6%)	26 (22.8%)	18 (16.2%)	0.21	
Muscular-/skeletal	54 (24%)	45 (39.5%)	9 (8.1%)	<0.001	
WEL-SF sum score	59.6 (16.1)	53.5 (16.2)	65.9 (13.3)	<0.001	
GSE sum score	31.3 (4.4)	30.7 (4.2)	31.9 (4.5)	0.04	
SEPA sum score	54.5 (14.2)	52.3 (13.7)	56.8 (14.4)	0.01	
IWQoL-lite sum score	67.9 (26.3)	47.9 (20.2)	88.5 (13.3)	<0.001	
SF-36 PCS score	45.1 (11.7)	39.9 (8.6)	53.5 (7.8)	<0.001	
SF-36 MCS score	46.2 (11.4)	40.7 (10.7)	51.9 (9.3)	<0.001	
Notes.

BMI Body Mass Index

WEL-SF Weight Efficacy Lifestyle Questionnaire Short Form

SEPA Self-efficacy for Physical Activity Scale

GSE General Self-efficacy Scale

IWQoL-Lite Impact of Weight on Quality of Life Lite Questionnaire

SF36 PCS and MCS Short Form 36 Physical- and Mental component summary

a p for group differences between non-operated and operated samples.

All values in mean, (SD) = standard deviation and (%).

Face validity of the WEL-SF

To be able to compare the results from the original WEL-SF with the psychometric properties obtained in the translated version, it is of major concern that the item construction in the two versions is semantically equivalent. Banduras’ test-theoretical approach to the development of self-efficacy scales worked as a guide during the item evaluation. We aimed to take the reader’s perspective using everyday vocabulary jargon. Furthermore, we aimed to avoid ambiguous or multi-barreled items that include different types of attainments within the same item, where the respondents may have different levels of perceived efficacy. Item 4 in the American WEL-SF (I can resist overeating when I am watching TV (or use the computer) may, in a Norwegian context, represent a double-wording problem in which it refers to disparate situations challenging eating self-efficacy. To assess our assumptions toward this potentially double-wording problem, we extracted the PC- item into a new item 9: “I can resist eating too much when I am using my PC/Ipad” and placed it elsewhere in the questionnaire. The mean score for the TV-item in the non-operated group was 6.07 and in comparison 8.90 for the PC-item. The difference was respectively the same in the operated group. The respondents seemed to experience significantly less eating efficacy while watching TV than when using the computer. As most respondents were checking the same, high response point on the PC-item, followed by a ceiling effect and low variability, it did not add relevant clinical information. We therefore decided to eliminate the PC-item from the questionnaire and maintained the original item. We worded the item closer to the original global WEL: “I can resist eating too much when I am watching TV”.

The translation process revealed divergence in translation of the concept “overeat”. The American “overeat” can qualify as a medical diagnosis (F50.4. ICD-10) within the broader framework of eating disorders (F50. ICD-10). Culturally and semantically “overeat” was interpreted as closer to the Norwegian “eating too much”. The Norwegian “overeating” seems as such conceptually more related to the American “binge-eating” which involves a pathological pattern of compulsive food intake. As we do not assume that all bariatric patients suffer from an eating disorder, we chose to reformulate “overeating” into “eating too much”. By this reformulation we also aimed to avoid potential stigmatizing and biases.

A pretest was performed to assess face validity and feasibility. Nine patients were recruited from outpatient consultations for this purpose one year after undergoing bariatric surgery. They were asked to complete the questionnaire and thereafter express whether the questionnaire was clear and easy to understand, covering topics of interest and if any items had been difficult to answer. In addition they were asked whether the questions were relevant for their situation. The pretest presented no corrections to the items and confirmed their clearness and relevance. Some of the respondents considered the introduction-text inappropriately long and redundant. We shortened and simplified the introduction accordingly. The participants in the pretest were not included in the psychometrical test performance of the translated version of the WEL-SF.

Reliability, internal consistency and sensitivity of WEL-SF

Cronbach’s alpha coefficients were 0.92 for the whole sample, 0.89 for the non-operated sample and 0.92 for the operated sample (Table 2). The Alpha value remained high (0.86–0.92) if single items were deleted (Table 3).

Table 2 Values for the Weight Efficacy Lifestyle Questionnaire Short Form (WEL-SF).

Morbidly obese patients (N = 225).

WEL-SF	Ceiling effect	Floor effect	Cronbach’s alpha	
	% max	% min		
All responders (N = 225)	8.9	0	0.92	
Non operated (N = 114)	0.9	0	0.89	
Operated (N = 111)	17.1	0	0.92	

Table 3 Mean, Standard deviation (SD) and Cronbach’s alpha if item deleted in the Norwegian version of the WEL-SF (N = 225).

	All patients	Non-operated	Operated	
	(N = 225)	(N = 114)	(N = 111)	
Item	Mean	SD	Alpha	Mean	SD	Alpha	Mean	SD	Alpha	
1. I can resist eating too much when I am anxious or nervous.	7.48	2.52	0.91	6.67	2.69	0.88	8.31	2.03	0.91	
2. I can resist eating too much on the weekend.	6.77	2.60	0.90	6.04	2.62	0.87	7.53	2.36	0.90	
3. I can resist eating too much when I am tired.	7.89	2.46	0.90	7.28	2.63	0.87	8.51	2.11	0.91	
4. I can resist eating too much when I am watching TV.	7.11	2.51	0.91	6.17	2.59	0.88	8.08	2.03	0.91	
5. I can resist eating too much when I am depressed or down	6.78	2.85	0.90	5.91	3.05	0.86	7.67	2.33	0.91	
6. I can resist eating too much when I am in a social setting or at a party.	7.44	2.50	0.91	6.83	2.69	0.88	8.06	2.13	0.92	
7. I can resist eating too much when I am angry or irritable.	7.72	2.36	0.90	6.90	2.59	0.87	8.57	1.74	0.91	
8. I can resist eating too much when others are pressuring me to eat.	8.37	2.44	0.91	7.67	2.75	0.88	9.09	1.82	0.91	

Construct validity and factor analysis

The data met the Kaiser–Meyer–Olkin measure (0.89) and the Bartlett’s test of spherity criterion (p < 0.001) for performing PCA. Following the Kaiser’s criterion, components with an eigenvalue > 1.0 were contained. The PCA was performed on the entire sample (n = 225) and the eight items of the WEL-SF loaded on one component only (Table 4) with an eigenvalue of 5.04 explaining 63% of the total variance. When performing the PCA on each of the subsamples this picture did not change. The 1 component solution had an eigenvalue (explained variance) of respectively 4.5 (56.4%) and 5.2 (65%) for the non-operated and operated sample. In comparison, Ames’ one-component solution accounted for 49% of the variance.

Table 4 Factor analysis results.

Comparison between reported one-component solutions in the samples. Morbidly obese patients (N = 225).

WEL-SF item text	Factor loading	
	Component 1	
	All patients	Non-operated	Operated	
	(N = 225)	(N = 114)	(N = 111)	
1. When I am anxious or nervous	0.77	0.73	0.77	
2. On weekends	0.85	0.81	0.89	
3. When I am tired	0.80	0.77	0.82	
4. When I am watching TV	0.78	0.71	0.79	
5. When I am depressed or down	0.85	0.84	0.84	
6. When I am in a social setting or party	0.71	0.66	0.73	
7. When I am angry or irritable	0.83	0.80	0.82	
8. When others are pressuring me to eat	0.74	0.68	0.77	
Notes.

Total variance explained: 63.0% (All patients), 56.4% (non-operated), 64.7% (operated).

Convergent validity

The correlation matrix for the sum score of the WEL-SF and the validating instruments covering our sample is illustrated in Table 5. The correlations ranged from .34 to .45 for all patients which represents a medium strong correlation (Cohen, 1988). Separating the groups, the correlations ranged from .12 to .37 in the non-operated group and .08 to .30 in the operated group, i.e., non to moderate strong correlations (Table 5).

Table 5 Correlation between Weight Efficacy Lifestyle Questionnaire Short Form and other measures (N = 225).

Variables	Pearson (r)	
	All patients (N = 225)	Non-operated (N = 114)	Operated (N = 111)	
SEPA	0.37 (p < 0.001)	0.37 (p < 0.001)	0.30 (p < 0.001)	
GSE	0.30 (p < 0.001)	0.29 (p = 0.002)	0.25 (p = 0.008)	
IWQoL-lite	0.45 (p < 0.001)	0.25 (p = 0.008)	0.27 (p = 0.004)	
SF36 (MCS)	0.40 (p < 0.001)	0.26 (p = 0.005)	0.26 (p = 0.006)	
SF36 (PCS)	0.34 (p < 0.001)	0.12 (p = 0.191)	0.08 (p = 0.427)	
BMI	−0.39 (p < 0.001)	−0.20 (p = 0.034)	−0.10 (p = 0.162)	
Change in BMI	NA	NA	−0.22 (p = 0.026)	
Notes.

WEL-SF Weight Efficacy Lifestyle Questionnaire Short Form

SEPA Self-efficacy for Physical Activity Scale

GSE General Self-efficacy Scale

IWQoL-Lite Impact of Weight Quality of Life Lite Questionnaire

SF36 Short Form 36 Scale

MCS Mental Composite Score

PCS Physical Composite Score

NA Not applicable

Sensitivity

WEL-SF sum score was lower in the non-operated than in the operated group in unadjusted analysis (Table 1). This difference remained using multiple regression as the WEL-SF sum score was 12.55 (95% CI: −16.59, 8.51) points lower in the non-operated than in the operated group (p < 0.001) (not shown).

Discussion

In this project we have translated and adapted the WEL-SF to Norwegian conditions, and tested its psychometrical properties in a population of morbidly obese patients accepted for bariatric surgery. During the translation and adaption process we discovered a few conceptual differences that were due to semantic or cultural conditions. The psychometric assessment of the final Norwegian version was consistent with those from the original WEL-SF in terms of internal consistency and data quality (Ames et al., 2012).

The structural validity of the translated WEL-SF was high, and the items all loaded on one component as suggested by Ames et al. (2012). Deciding how many factors to retain is a critical component of exploratory factor analysis, and the one component solution remained when performing PCA on the two subsamples. There is no clear consensus concerning sample size requirements for factor analysis (Williams, Brown & Onsman, 2010), but even though the present study was based on samples less than 200 subjects (Kline, 2000) we consider the results indicative of a structural robustness.

We obtained high item to sum score correlations calculated for both the entire sample and the subsamples indicating that the instrument measures one underlying construct (Streiner & Norman, 2008). A high overall reliability coefficient and corresponding alpha values if single items were deleted provides further evidence in support of the construct validity.

The WEL-SF sum score was correlated with SEPA and GSE sum scores in the subsamples as well as for the entire sample, where patients who reported high levels of efficacy expectations toward eating behavior also tended to present high levels of confidence toward physical activity and challenging obstacles in general. The association between eating efficacy and physical activity expectations is in correspondence with earlier findings (Morin, Turcotte & Perreault, 2013; Sallis et al., 1988; Wing et al., 2001), and was equally pronounced in the subsamples as in the entire sample. We also found an association between the WEL-SF and the IWQOL-Lite in both subsamples. Patients reporting high levels of confidence toward eating behavior seem to experience higher quality of life in spite of their obesity. Furthermore, we measured the correlations between WEL-SF and the SF36’s two subdomains: mental and physical composite scores. The obtained association between eating efficacy and the mental domain were significantly correlated for all the samples. This corresponds with Ames remark during the item selection for the WEL-SF, where the highest loaded items on the component “confidence in ability to resist eating”, appeared to represent negative emotions (Ames et al., 2012). Former studies have emphasized the association between emotional eating and poor weight loss maintenance (Niemeier et al., 2007; Phelan et al., 2009). From this we might deduce that highly reported eating efficacy expectations may be connected to personal skills and strategies for managing emotional eating situations. A significant correlation also appeared between WEL-SF and the SF36’s physical domain accounting for the entire sample. We did not, however, find any significant associations between WEL-SF and the SF36’s physical domain in the two subsamples. Overall the correlations were largest in the analysis using the whole sample, probably reflecting that the variation in scores was greater in this group.

Strength and limitations

A cross-sectional design represents potential limitations due to its lack of time measurement (Polit & Beck, 2008). Nevertheless, we find this methodological approach appropriate for the present study due to our intention of inferring WEL-SF’s present psychometrical properties for future predictive purposes.

Data from both non-operated and operated patients strengthened the study in terms of a larger sample-size, and by bringing the opportunity to assess the WEL-SF’s sensitivity for the overall different eating pattern between the two groups. The subgroups were similar regarding socio-demographic variables, but had different health profiles, as expected. This was, nevertheless, taken into account by conducting the statistical analysis for the two subsamples in addition to the entire sample to visualize the outcome differences and similarities.

We noted some possible problems with ceiling effect in the operated group. This may be a problem if the WEL-SF is to be used for measuring change over time because of potentially low responsiveness beyond one year after surgery. Studies with longer follow-up should be performed in order to explore this issue, and caution must be taken in future studies if ceiling effects are common in Norwegian bariatric patients.

The response bias (Polit & Beck, 2008) was reduced due to the consecutively and convenient sampling procedure, contributing to a very high response rate and only one missing value. The referral of patients to the hospital from general physicians throughout the country strengthens the representativeness and generalizability of the results. Nevertheless, as most of the respondents were Caucasian, all admitted for surgery, the results may not be valid for obese patients from other ethnic groups, or for those seeking non-surgical treatment.

Conclusion

With the present study, a Norwegian version of the WEL-SF is made available for use for clinical work and research assessing eating self-efficacy in morbidly obese patients eligible for bariatric surgery. Morbidly obese patients not seeking bariatric surgery should be addressed in future studies to increase the utility of the WEL-SF in a Norwegian population.

Supplemental Information

Supplemental Information 1 American and Norwegian version of the WEL-SF

Click here for additional data file.

Supplemental Information 2 Raw data and explanations

Click here for additional data file.

Supplemental Information 3 Code-book

Click here for additional data file.

We thank the clinical experts Hanne Rosendahl Gjessing (Clinical Nutritionist and Fellow Researcher, University of Bergen), Camilla Laukeland (Clinical Nutritionist, Førde Central Hospital), Hilde Blindheim Børve (Health Educator, Haukeland University Hospital) and Kim Waardal (MD, Haukeland University Hospital) for their contribution to the translation- and adaptation process, and the patients for completing the questionnaires. We also thank Professor Simon Øverland (Norwegian Institute of Public Health) for proofreading the manuscript and MD/PhD Villy Våge (Voss Hospital, Haukeland University Hospital) for final adjustments. The authors are grateful to the management of the local hospital making time and locations available so that this study was possible.

Additional Information and Declarations

Competing Interests

Author Contributions

Human Ethics

The authors declare there are no competing interests.

Tone N. Flølo, John R. Andersen, Hans J. Nielsen and Gerd K. Natvig conceived and designed the experiments, performed the experiments, analyzed the data, contributed reagents/materials/analysis tools, wrote the paper, prepared figures and/or tables, reviewed drafts of the paper.

The following information was supplied relating to ethical approvals (i.e., approving body and any reference numbers):

The study was approved by the Regional Committee for Medical Research Ethics in Western Norway and performed in accordance with the Helsinki Declaration (Saksnr 2012/1481).

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
