# Peer review of "Translation, adaptation, validation and performance of the American Weight Efficacy Lifestyle Questionnaire Short Form (WEL-SF) to a Norwegian version: a cross-sectional study"

_PeerJ, doi:10.7717/peerj.565_

## Round 0.1 · original submission · Minor Revisions

· Academic Editor

Minor Revisions

Please revise your manuscript in response to the reviewer suggestions, and respond to each point that has been raised.

Reviewer 1 ·

Basic reporting

No comments

Experimental design

Since the study has two aims and a four part research question, it would be helpful in the methods section to identify which question each section is designed to address. For example, construct validity and factor analysis section (study aim 2 part c) of something similar to help with organization and flow.

Validity of the findings

Analyses address the proposed aims and study questions adequately. Tables are well organized and informative.

Please clarify the response rate of 100%. Does this mean that 114 consecutive patients agreed to participation before undergoing the operation and that 111 postoperative patients agreed to study participation? How were missing data handled?

Comments for the author

Literature review is thorough. The study purpose and research questions are clearly stated.

The discussion is well-written. Limitations of cross-sectional measurement are addressed. No definitive conclusions regarding improvement in eating self-efficacy from pre to post weight loss surgery can be derived from this study. However, this is an acceptable limitation given the primary aim of translating the American English language WEL-SF to Norwegian and to provide preliminary validation data on the translated version of the questionnaire. This study also addresses limitations of the American English Language WEL-SF that should be considered in future validation studies. For example, in item 4 the double-wording problem was addressed. In this sample, the authors found that adding a PC-item was clinically irrelevant.

---

## Round 0.2 · accepted · Accept

· Academic Editor

Accept

Based upon your satisfactory response to the reviewers' comments, your paper is now suitable for publication in PeerJ.